# The relationship between plant-eating and hair evacuation in snow leopards (*Panthera uncia*)

Hiroto Yoshimura[1]*, Huiyuan Qi[1], Dale M. Kikuchi[2], Yukiko Matsui[3],
Kazuya Fukushima[4], Sai Kudo[4], Kazuyuki Ban[5¤], Keisuke Kusano[6], Daisuke Nagano[6],
Mami Hara[7], Yasuhiro Sato[7], Kiyoko Takatsu[3], Satoshi Hirata[1], Kodzue Kinoshita[1]*

1 Wildlife Research Center, Kyoto University, Kyoto, Japan, 2 Department of Mechanical Engineering Science, Tokyo Institute of Technology, Tokyo, Japan, 3 Tama Zoological Park, Hino, Tokyo, Japan, 4 Sapporo Maruyama Zoo, Sapporo, Hokkaido, Japan, 5 Omuta city zoo, Omuta, Fukuoka, Japan, 6 Kumamoto City Zoological and Botanical Gardens, Kumamoto, Japan, 7 Nagoya Higashiyama Zoo and Botanical Gardens, Nagoya, Aichi, Japan

¤ Current address: Morioka Zoological Park, Morioka, Iwate, Japan
* hrt.yoshimura@gmail.com (HY); kinoshita.kodzue.8v@kyoto-u.ac.jp (KK)

**Data Availability Statement:** All relevant data are within the paper and its Supporting Information files.

**Funding:** Leading Graduate Program in Primatology and Wildlife Science (http://www.

## Abstract

Although most felids have an exclusive carnivore diet, the presence of plant matter in scat has been reported among various species. This indicates that there may be an adaptive significance to the conservation of plant-eating behavior in felid evolution. Some studies have hypothesized that felids consume plants for self-medication or as a source of nutrition. In addition, it is thought that plant intake helps them to excrete hairballs, however, no scientific work has confirmed these effects. Thus, the objective of this study is to investigate the relationship between plant intake and hair evacuation in felid species. We selected snow leopards (*Panthera uncia*) as the study species because they have longer and denser hair than other felids. The behavior of 11 captive snow leopards was observed and scat samples from eight of them and two other captive individuals were analyzed. Snow leopards evacuate hair possibly by vomiting and excreting in scats. The frequency of plant-eating and vomiting and the amount of hair and plant in scat were evaluated. We found that the frequency of vomiting was much lower than the frequency of plant-eating. In addition, there was no significant relationship between the amount of plant matter contained in scats and the amount of hair in scats. Contrary to the common assumption, our results indicate that plant intake has little effect on hair evacuation in felid species.

## Introduction

Ingestion of plants is one of the mysteries in felid ecology because of their inability to properly digest plant matter [1,2]. Although felids are strict carnivores [3], plant ingestion is a conserved behavioral trait among felid species, which has been observed in both captive and free-ranging conditions [4–9]. Moreover, empirical studies showed that cellulose, a structural component

wildlife-science.org/index-en.html) supported logistic cost for the field work to HY. This work was supported by JSPS KAKENHI Grant-in-Aid for Scientific Research (B) (20H03008) to KK. The funders had no role in study design, data collection and analysis, decision to publish, or preparation of the manuscript.

**Competing interests:** The authors have declared that no competing interests exist.

of plant cell walls, reduced the digestibility of dry matter in Amur leopard cat (*Prionailurus bengalensis euptilura*) and Turkmenistan caracal (*Caracal caracal michaelis*) [10] and energy in domestic cat (*Felis silverstris catus*) [11]. Given that plant ingestion is biologically costly, there should be an adaptive significance to this behavior that compensates for the potential costs. However, no reports have investigated the adaptive significance of plant ingestion in felids.

There are three hypotheses that may explain the adaptive significance of plant ingestion in felids. First, a study on leopard cats (*Prionailurus bengalensis*) suggests the food source hypothesis, which proposes that plants have some nutritional value for them [12]. This hypothesis is based on the fact that scat samples of leopard cats in southwest China contained DNA of *Solanum* and *Rosoideae* species that produce berry fruits that are rich in sugar and nutrients [12]. However, the food source hypothesis may not always apply to felids, as they frequently consume grass or leaves [7,13,14], which should be less nutritional than fruits. Second is the self-medication hypothesis: Hart [15] suggests that dogs use plant materials for self-medication to expel parasites or treat inflammation; this was particularly common in young individuals, which are less immune to intestinal parasites. However, this self-medication effect has not been verified in dogs or cats. The third hypothesis suggests that plant intake is related to hair evacuation [16]: hairballs prevent digestion and they must not fill the digestive tract [17]. Felines often ingest their own hair while grooming, as well as the hair of their prey items while eating. Felids evacuate hair through vomiting or expelling as scat. Although it is thought that ingested plants aid in excreting hairballs [18], there is still no empirical evidence of this. Therefore, the hair evacuation hypothesis has yet to be tested conclusively.

Snow leopards (*Panthera uncia*) is an endangered cat species that live in high altitude regions (1,220 to > 5,000 m) of Central Asia [19], designated as VU (vulnerable) in the IUCN Red List of Endangered Species [20]. Bharal (*Pseudois nayaur*) and Siberian ibex (*Capra sibirica*) are primary prey species of snow leopards [21]; their ranges almost entirely overlap with that of snow leopards [22]. Large portions of snow leopards' natural habitat are devoid of tree cover, given the predominance of alpine and desertic climate conditions in their natural range. The vegetation in their range varies from scrubland and desert to forest-alpine ecotones [21]. The presence of plant material in snow leopard scat has been reported in several research areas, despite the relatively low abundance of vegetation across their habitat [23–28]. For example, it was reported that plant materials occurred in 62% of scat samples collected in Phu valley, Nepal. In some cases, scat content consisted almost entirely of plants [29]. Snow leopards also have longer and denser hair than other felids as an adaptation to life at high altitude [30], which indicates a relatively high frequency of hair ingestion through grooming and thus, a correspondingly frequent evacuation. Based on the above characteristics, we chose snow leopards as a suitable felid species for which to investigate the effects of plant ingestion and hair evacuation.

In this study, we tested the hair evacuation hypothesis in captive snow leopards, through behavioral observations and scat analysis. Behavioral observations examined the frequency of plant ingestion and vomiting to identify the potential effect of plant intake on vomiting. We collected scat samples and measured the amount of plants and hair and examined their statistical relationship. Together, these analyses provide quantitative evidence to test the hair evacuation hypothesis.

## Methods

Management of the captive snow leopards in this study followed the Code of Ethics of the Japanese Association of Zoos and Aquariums. Sampling procedures were noninvasive and

approved by each zoos and Animal Experimentation Committee of Wildlife Research Center of Kyoto University. This study complied with applicable national laws.

## Animals

The subject animals were 13 snow leopards (7 females, 6 males) kept in zoos in Japan. All individuals were housed separately when inside. Basically, six snow leopards (Female 1–3, Male 1-3) at Tama Zoological Park used two outdoor enclosures one by one, although Female 3 and Male 3 (mother and cub) used the enclosure at the same time in 2018 only. Female 4 and Male 4 at Kobe Oji Zoo used the outdoor enclosure at the same time. Female 5 and Male 5 at Sapporo Maruyama Zoo, Female 7 and Male 6 at Nagoya Higashiyama Zoo and Botanical Gardens also used the same enclosures one at a time. Information about the animals is presented in Table 1. Depending on the zoo, all snow leopards were fed mainly horseflesh, chicken breast meat, and/or chicken bone. In general, their food was provided every day except on weekly fasting days. At Tama Zoological Park, snow leopards were fed whole rabbits once a week, and a bundle of straw for play was given to them on three days during the observation period in 2019. Plants were not provided as food at any zoo, however, all individuals except Female 7 and Male 6 were able to access plants in outside enclosures for at least one hour every day.

## Behavioral observation and scat analysis

The behavioral observation was conducted on 11 individuals. Focal animals were observed directly and/or on video while they were in the outside enclosure and their behavior was continuously recorded. Behavior (move, rest, search, marking, plant eat, play, other) was recorded based on Freeman 1974 [31]. Plant-eating behavior was recorded only when it was certain that they had plants that were growing in the enclosures in their mouths and bite wooden structures (e.g., benches). Since their behavior appears to be unusual during scorching weather or heavy rain, the data from approximately 19 hours (4.5% of total observation) was excluded from the analysis. Behavior bouts were defined as the same bout if a behavior was resumed in 30 seconds and no other behavior (except for move, stand, and rest) was observed. The frequency of plant-eating of each individual was calculated as bouts/hour during each observation period.

**Table 1. Focal invidivual characteristics.**

| Individual ID | Name | Sex | Age at study | Location |
|---|---|---|---|---|
| Female 1 | Asahi | Female | 8 | Tama Zoological Park |
| Female 2 | Mirucha | Female | 11 | |
| Female 3 | Mimi | Female | 10 | |
| Male 1 | Valdemar | Male | 14 | |
| Male 2 | Kovo | Male | 5 | |
| Male 3 | Fuku | Male | 1 | |
| Female 4 | Yukko | Female | 10 | Kobe Oji zoo |
| Male 4 | Fubuki | Male | 2 | |
| Female 5 | Sizim | Female | 9 | Sapporo Maruyama zoo |
| Male 5 | Akbar | Male | 14 | |
| Female 6 | Supica | Female | 14 | Omuta city zoo / Kumamoto City Zoological and Botanical Gardens |
| Female 7 | Rian | Female | 9 | Nagoya Higashiyama Zoo and Botanical Gardens |
| Male 6 | Yukichi | Male | 10 | |

Age at study is age at the time of the latest study.

The collection of scat samples was conducted for 10 individuals (Male 1, 2, and 3 were excluded). Whole scats were basically collected every day during the sampling period and stored in airtight plastic bags at -20 degrees until analysis. Scat excreted at one time was treated as one sample. Each sampling period was 10 to 14 days in a row. The behavioral observation was conducted simultaneously when possible, to distinguish scat samples and determine the order if several individuals used the same enclosure in one day or if they defecate more than once in the outside enclosure.

When it was not apparent which individual a scat was from at Kobe Oji Zoo or Nagoya Higashiyama Zoo and Botanical Gardens, we used fecal DNA to identify the sex because individuals using the same enclosure were different sexes. DNA analysis used primers from Sugimoto *et al.* 2006 [32]. The same procedure was repeated three times and the sex was determined only when the result was consistent.

The scat samples were freeze-dried overnight (FDU-1200, EYLA, Tokyo), then weighed, and 0.10–0.50 g of powdery parts from each dried fecal sample were removed for other analysis if possible. The rest of the sample was then washed in tap water with 1 mm mesh to pick out undigested matters, hair, plants including pieces of wooden benches and other material (e.g., gravel). The contents from each sample were packed in airtight plastic bags, then freeze-dried overnight and weighed. Scat samples collected in Tama Zoological Park that included rabbit hair, bone or straw were excluded from the analysis to minimize the differences between captive conditions. When deciding the order of scat samples, samples lighter than 5 g in dried weight (DW) were not counted because at times the same individual defecated several times within a few hours and the small samples were considered as a portion of a larger scat sample.

## Data analysis

Data were analyzed using Microsoft Excel (Microsoft, Tokyo), and R software (version 3.6.1., R foundation for Statistical Computing 2019) [33].

To test the quantitative relationship among scat sample contents, the amount of plant matter in a scat sample, the amount of plant matter contained in the scat sample evacuated before hair was excreted, and the amount of plant matter contained in the scat sample evacuated after hair was excreted were set as fixed effect ("s-plant", "b-plant", "a-plant") and the amount of hair in the scat sample was set as the objective variable ("hair"). The amounts of plant matter contained in scat samples evacuated before or after hair was excreted were added as variables to consider the possibility that the transition rate of the plant materials and hair could be different. The gamma distribution was selected because the objective variable was continuous and should not have a negative value. Thus, a generalized linear model (GLM) and a generalized linear mixed model (GLMM) with gamma distribution and identity link function was applied. Sampling unit was set as random effect ("individual_period") in GLMMs. Each of the three variables was applied one by one (Table 4, Table A in S1 File), resulting in three one-variable models. To use gamma distribution, when the amount of hair was zero, the value was replaced with 0.0001 based on the roundoff error 0.0005 [34,35] in six samples.

Bayesian estimation by "rstanarm" package version 2.19.2 was used to estimate the coefficients of the models. Normal distribution with a mean of zero was used prior because either a positive or negative estimate was allowed, and the sample sizes were small. We ran four independent Markov chains of each model. All iterations were set to 5,000 and the burn in samples were set to 2,500. The value of Rhat for all parameters was equal to 1.0, indicating convergence across the four chains [36,37]. We concluded that the estimate was significantly different from zero if the 95% CI range did not stride over zero.

## Results

### Plant-eating and vomiting

Behavioral observation was conducted for a total of 417 hours from September 2018 to October 2019, and 398 hours were used for the analysis. Plant-eating behavior was observed in 10 out of 11 individuals. This behavior was most frequent in Male 3 (1.19 bouts/hour) and least frequent in Female 6 (0.06 bouts/hour) (Table 2). The longest bout continued for 6 minutes 55 seconds (Male 3), while the shortest bout was only 2 seconds (Male 4). In each individual, these plant-eating behaviors were observed on several days. Vomiting was observed just once in Female 2 and Female 3 and twice in Male 3.

### Plant-eating and hair evacuation in scat samples

In total 192 scat samples were collected from 8 individuals that were kept in enclosures with plants. Three samples from Tama Zoological Park were not used because they were mixture of several scats. Samples that were lighter than 5 g in dried weight were excluded ($n = 17$). Dried weight data was not available for five of the scat samples from Female 5; however, three samples were still included because total dried weight of scat contents was over 5 g. Live prey (rabbit) and straw were only provided at Tama Zoological Park, so samples that contained rabbit bones and hair or straw were also excluded ($n = 23$). This left 147 samples that were included in the analysis. Of the 147 samples, 141 samples (96%) contained snow leopard hair and 95 samples (65%) contained plant matter. As shown in Fig 1, plants were evacuated in an undigested state. Additionally, 14 scat samples from Female 7 and 15 samples from Male 6 were collected. These two were kept in enclosures without plants but four samples from Male 6 contained pieces of wood from the wooden bench. One sample from Female 7 that was lighter than 5 g was excluded. The results of sex identification were not consistent in three samples thus we didn't use them in the study. Dried weight of scat, and the amounts of hair and plants included in scat samples are presented in Table 3.

The scatter plot shows the relationship between the amounts of hair and plant included in scat samples (Fig 2). Hairs were evacuated in scat samples regardless of the presence or absence

**Table 2. Plant-eating and vomiting behavior of snow leopards observed in zoos in Japan.** Mean bout length of plant-eating with standard deviation (±SD).

| Location | Individual | Observation period | plant eat (bout) | Vomit (bout) | observed time (h) | Frequency of plant eat (bout/h) | Bout length of plant-eating (min) |
|---|---|---|---|---|---|---|---|
| Tama Zoological Park | Female 1 | Sep 22, 2018 to Dec 2, 2018 | 23 | 0 | 22.9 | 1.00 | 1.0 ± 0.9 |
| | Female 2 | | 3 | 0 | 26.5 | 0.11 | 0.7 ± 0.7 |
| | Female 3 | | 14 | 0 | 38.8 | 0.36 | 0.3 ± 0.2 |
| | Female 1 | Oct 4, 2019 to Oct 16, 2019 | 15 | 0 | 19.8 | 0.76 | 0.7 ± 0.6 |
| | Female 2 | | 10 | 1 | 15.5 | 0.65 | 0.4 ± 0.6 |
| | Female 3 | | 10 | 1 | 26.5 | 0.38 | 0.5 ± 0.6 |
| | Male 1 | Sep 22, 2018 to Dec 2, 2018 | 0 | 0 | 8.8 | 0.00 | - |
| | Male 2 | | 25 | 0 | 30.4 | 0.82 | 0.9 ± 0.8 |
| | Male 3 | | 29 | 2 | 24.3 | 1.19 | 1.0 ± 1.5 |
| Kobe Oji Zoo | Female 4 | May 22, 2019 to Jun 4, 2019. | 8 | 0 | 105.7 | 0.08 | 1.7 ± 1.6 |
| | Male 4 | | 9 | 0 | 105.3 | 0.09 | 0.7 ± 0.5 |
| Sapporo Maruyama Zoo | Female 5 | Sep 10, 2019 to Sep 23, 2019. | 10 | 0 | 21.6 | 0.46 | 1.1 ± 1.1 |
| | Male 5 | | 10 | 0 | 20.8 | 0.48 | 0.9 ± 0.8 |
| Kumamoto City Zoological and Botanical Gardens | Female 6 | Aug 1, 2019 to Aug 14, 2019. | 2 | 0 | 36.3 | 0.06 | 1.4 ± 0.2 |

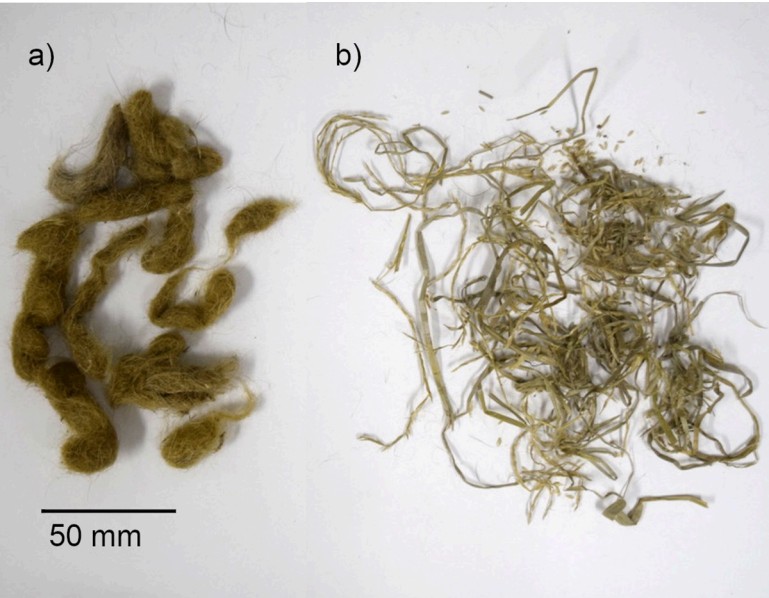

**Fig 1. Hair and plant collected from snow leopard scat sample.** Example of a) hair and b) plants collected from one scat sample.

of plants. Following analyses were conducted on individuals, except Female 7 because there were no plants growing in the enclosures at Nagoya Higashiyama Zoo and Botanical Gardens and no plant was collected from its scats. In either dried weight of hairs or plants, significant difference was obtained among sampling units (i.e. individuals and/or sampling periods) (hair; $p < 0.001$, plant; $p < 0.001$, respectively, Kruskal–Wallis rank sum test). Thus, we set

**Table 3. The mean total hair and plant amounts (g, dried weight) contained in scat samples (mean ± SD).**

| Location | Individual | Sampling period | Sampling unit | Hair in scat (g) | Plant in scat (g) | Scat weight (g) |
|---|---|---|---|---|---|---|
| Tama Zoological Park | Female 1 | Oct 4, 2019 to Oct 16, 2019. | f1 ($n = 8$) | 0.53 ± 0.47 | 0.10 ± 0.11 | 37.67 ± 20.11 |
| | Female 2 | | f2 ($n = 7$) | 1.19 ± 2.49 | 0.03 ± 0.04 | 26.72 ± 19.83 |
| | Female 3 | | f3 ($n = 11$) | 2.41 ± 1.40 | 0.04 ± 0.02 | 22.17 ± 9.69 |
| | Male 1 | No data | | No data | No data | No data |
| | Male 2 | | | | | |
| | Male 3 | | | | | |
| Kobe Oji Zoo | Female 4 | Aug 24, 2018 to Sep 5, 2018 | f4_1 ($n = 12$) | 2.80 ± 2.12 | 0.35 ± 0.31 | 45.51 ± 19.12 |
| | | May 22, 2019 to Jun 4, 2019. | f4_2 ($n = 18$) | 2.30 ± 1.65 | 0.05 ± 0.12 | 50.17 ± 29.59 |
| | | Aug 26, 2019 to Sep 4, 2019. | f4_3 ($n = 13$) | 1.45 ± 1.15 | 0.05 ± 0.08 | 41.60 ± 16.95 |
| | Male 4 | May 22, 2019 to Jun 4, 2019. | m4_1 ($n = 11$) | 3.73 ± 3.47 | 0.03 ± 0.13 | 35.16 ± 11.51 |
| | | Aug 26, 2019 to Sep 4, 2019. | m4_2 ($n = 9$) | 1.20 ± 0.90 | 0.23 ± 0.26 | 34.31 ± 10.87 |
| Sapporo Maruyama Zoo | Female 5 | Sep 10, 2019 to Sep 23, 2019. | f5 ($n = 16$) | 1.28 ± 0.83 | 0.10 ± 0.12 | 27.05 ± 11.96* |
| | Male 5 | | m5 ($n = 17$) | 0.16 ± 0.14 | 0.14 ± 0.13 | 34.98 ± 18.54 |
| Omuta city zoo | Female 6 | Jul 20, 2018 to Aug 2, 2018. | f6_1 ($n = 13$) | 3.36 ± 1.95 | 0.12 ± 0.14 | 25.39 ± 6.59 |
| Kumamoto City Zoological and Botanical Gardens | Female 6 | Aug 1, 2019 to Aug 14, 2019. | f6_2 ($n = 12$) | 1.60 ± 0.75 | 0.07 ± 0.09 | 37.67 ± 14.31 |
| Nagoya Higashiyama Zoo and Botanical Gardens | Female 7 | Mar 2, 2020 to Mar 11, 2020 | f7 ($n = 13$) | 1.85 ± 1.61 | 0.00 ± 0.00 | 40.81 ± 23.34 |
| | Male 6 | | m6 ($n = 15$) | 1.41 ± 1.29 | 0.06 ± 0.14 | 34.76 ± 20.94 |

*Data was not available for three samples.

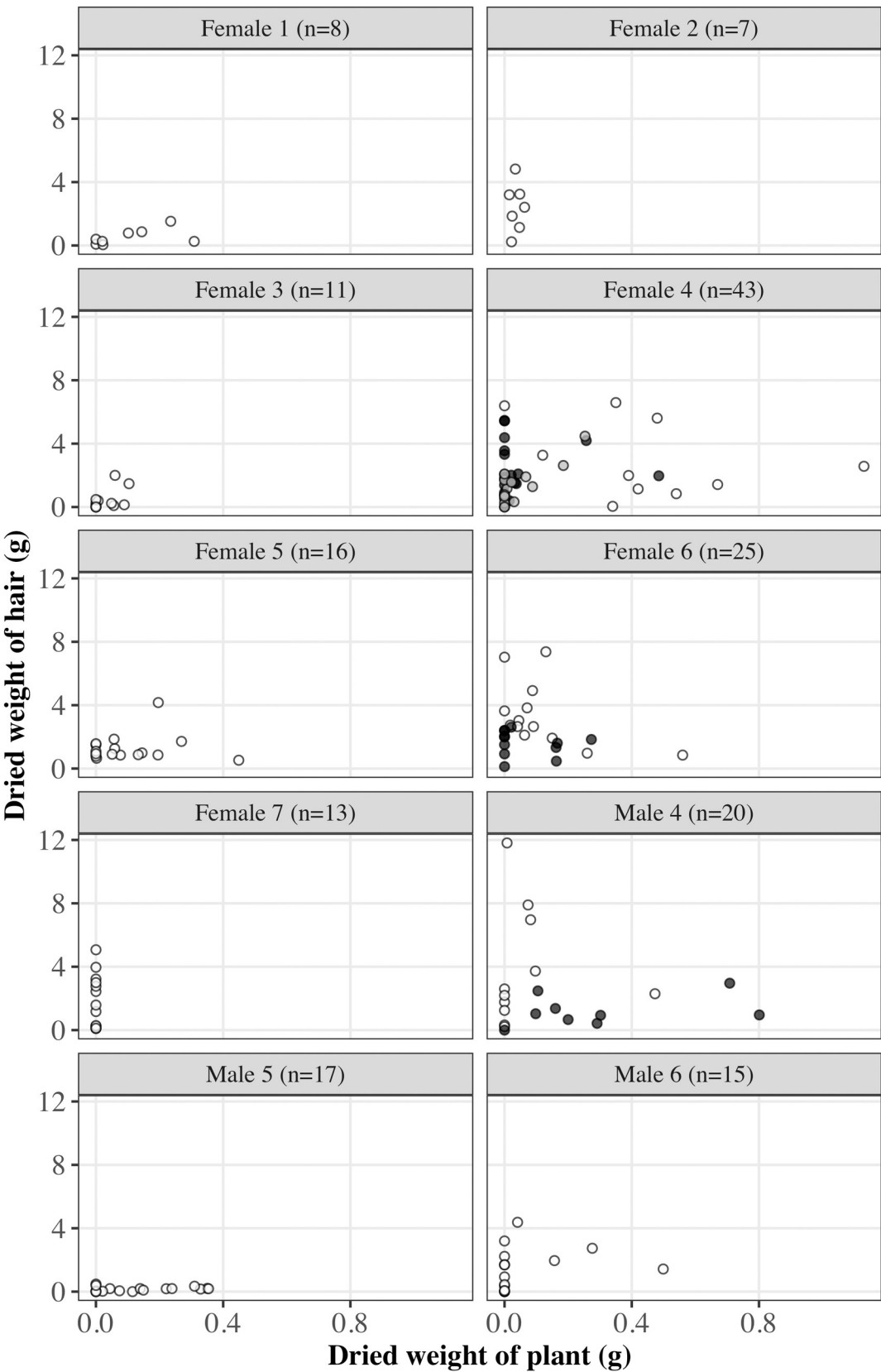

**Fig 2. Total dried weight of plant and hair contained in snow leopard scat samples.** White, black and gray represent the first, second, and third sampling period, respectively. For the details of the individuals, see Table 3.

**Table 4. The effect of plants on the amount of hair in the scat sample, as estimated by GLMM (Bayesian estimation).** Coefficients with SD.

| Models | Model 1 | Model 2 | Model 3 |
|---|---|---|---|
| Objective variable | The amount of hair in scat sample | | |
| Fixed effect | The amount of plant contained in the same sample | The amount of plant contained in the sample evacuated before hair was excreted | The amount of plant contained in the sample evacuated after hair was excreted |
| Estimate (± SD) | 0.685 ± 0.974 | 0.852 ± 0.901 | 0.673 ± 1.105 |
| 95% CI range | -0.970–2.869 | -0.639–2.938 | -1.142–3.255 |

"individual_period" as a random effect when creating the generalized linear mixed models (GLMM). Prior to modeling for the estimation of the quantitative relationship between amounts of plants and hairs evacuated, we removed scat samples if the defecation order of samples was not clear. When data of the previous or next scat sample was not available, they were also excluded. A total of 107 samples were used in this analysis. To consider the possibility that the transition rate of plant materials and hair will be different, three variables (the amount of plant contained in the same scat sample, and in the samples evacuated before and after hair was excreted) were set as fixed effects (Fig in S1 File). The estimated coefficients of fixed effects in each of the one-variable models are shown in Table 4. None of the three variables were significantly different from zero (Table 4). In order to find out if there was a sampling unit that had a relationship between hair and plant in the scat sample, we constructed generalized linear models for each sampling unit. However, irrespective of sampling unit, the estimated coefficients of three variables were not significantly different from zero (Table A in S1 File).

## Discussion

The results of this study confirm that captive snow leopards eat plants fairly frequently and this behavior did not induce vomiting. Together with findings from reports of plant containment in snow leopard scat [27,29], our results suggest that plant-eating is a normal behavior for this species, both in the wild and in captivity. Therefore, growing plants in captive snow leopards' enclosures might be more suitable to bring out their natural behavior in captivity, contributing to the enrichment and thus improving their welfare.

In scat samples, the amount of hair did not increase in relation to the amount of plants ingested and there was no quantitative relationship between them. Therefore, we conclude that ingested plants do not have an immediate function to evacuate hair. Our data clarified that the traditional hypothesis that ingested plants help to excrete hairballs is not necessarily credible.

In the wild, snow leopards move daily across long distances [21] (e.g., 12 km/day in Mongolia [38]). Their broad home range and rugged habitat make it difficult to observe their natural behavior and hinder our ability to consistently collect scat samples from the same individual. Studies on captive individuals have enabled detailed observations of this plant-eating behavior, allowing continuous sampling to test this relationship with a time difference. This study further demonstrates the importance of studying captive individuals to understand wildlife.

We observed plant-eating behavior in all but one of the snow leopards (Table 2), despite daily feeding for the study duration. Hoppe-Dominik [8] suggested that leopards (*Panthera pardus*) intentionally eat grass during periods of prolonged starvation to keep their digestive system functioning. However, our results indicate that plant-eating is also common in well-fed captive snow leopards. Contrary to the frequent plant-eating, we rarely observed vomiting.

The frequency was not consistent with that of plant-eating, thus we conclude that snow leopards did not eat plants to promote vomit hairballs through stimulation of the throat or stomach. According to an internet survey targeting the owners of plant-eating dogs, only 22% of the dogs frequently vomit after eating plant materials, thus they concluded that plant-eating is not related to vomiting [39].

In this study, hairs were evacuated in scats, regardless of the presence or absence of plants in the enclosures. Also, the amount of hair and plant in scat samples were varied among individuals and/or sampling terms, indicating individual differences and/or differences between sampling periods in the amount of hair and plants within the enclosures. The length of coat hair of snow leopards is reported to differ with the seasons [40]. Additionally, depending on the season and the zoo, the abundance and composition of vegetation within the enclosures appeared to vary. This might have caused the individual differences in hair and plant matter in the scat samples. When creating the GLMM, the difference between individual and/or sampling period was taken into consideration. Still, the results revealed that all three variables did not have significant effects on the amount of hair contained within a scat sample. Furthermore, the amount of plant matter in scat samples had no significant relationship with the amount of hair, regardless of time difference. Although we cannot rule out a causative relationship between the amount of plant in scat and hair evacuation, this study was the first to provide evidence that plant matter in scat had no quantitative effect on hair evacuation.

In this study, we used captive snow leopards to obtain continuous data to estimate the relationship between plant ingestion and hair evacuation over a period of time. However, captive animals might ingest much less hair from their diet than in the wild because they are mainly fed meat as opposed to live prey. Therefore, we should note that the effect of plant intake may be underestimated due to the lack of prey hair ingestion. Also, plant composition was different from their wild habitat. In some habitats, it was reported that many scats of snow leopards contain *Myricaria* [27], but snow leopards also intake other plant species including grasses in other habitats [41]. Although in this study we let snow leopards voluntarily select when and which plant to eat, there was a chance that the effect of plant intake was not detected because the plant species they ate in zoos did not have the required traits.

As stated in introduction, several factors are expected to make strict carnivores eat plants. Our study tested one hypothesis about the adaptive significance of plant-eating in strict carnivores for the first time. We confirmed that snow leopards voluntary and frequently eat plants. However, our results did not support the hair evacuation hypothesis, therefore the advantage of plant intake for snow leopards is still unclear. Further studies are required to evaluate the effects of plant intake not only on physical aspects but also on chemical aspects such as antibiotic compounds. Information about the plant species that snow leopards use in the wild may provide a novel hypothesis to be tested. Another area of research that requires further investigation would be to identify the driving factors of carnivore plant consumption in the wild. Although carnivores are known to be indifferent to sugars, as demonstrated by a study on domestic cats that revealed a lack of sweet taste receptors [42], other flavors (e.g., bitter taste), olfactory clues [43], or plant texture [8] might be influencing carnivore plant-eating behaviour. To truly understand their ecology, we should pay attention not only to the prey animals but also to the plant species present in the scat of strict carnivores.

## Supporting information

**S1 Dataset. Dataset of scat samples.**
(XLSX)

**S1 File. Fig and Table A.** This file includes scatter plot of hair and three variables (s-plant, b-plant, a-plant) and results of the GLM.
(DOCX)

**S2 File. Results of Bayesian estimation for GLM.**
(XLSX)

**S3 File. Results of Bayesian estimation for GLMM.**
(XLSX)

## Acknowledgments

The authors are grateful to the staffs of the Tama Zoological Park, Omuta city zoo, Kobe Oji Zoo, Sapporo Maruyama Zoo, Kumamoto City Zoological and Botanical Gardens and Nagoya Higashiyama Zoo and Botanical Gardens for their generous assistance. We are appreciated Prof. Naofumi Nakagawa and Assoc. Prof. Michio Nakamura of Laboratory of Human Evolution Studies, Kyoto University, Mr. Sota Inoue and members of Wildlife Research Center of Kyoto University for helpful discussion and advice. The authors would like to thank Enago (www.enago.jp) for the English language review.

## Author Contributions

**Conceptualization:** Hiroto Yoshimura.

**Data curation:** Hiroto Yoshimura.

**Formal analysis:** Hiroto Yoshimura.

**Investigation:** Hiroto Yoshimura, Huiyuan Qi.

**Project administration:** Hiroto Yoshimura, Kodzue Kinoshita.

**Resources:** Yukiko Matsui, Kazuya Fukushima, Sai Kudo, Kazuyuki Ban, Keisuke Kusano, Daisuke Nagano, Mami Hara, Yasuhiro Sato, Kiyoko Takatsu.

**Supervision:** Satoshi Hirata.

**Writing – original draft:** Hiroto Yoshimura, Huiyuan Qi, Kodzue Kinoshita.

**Writing – review & editing:** Dale M. Kikuchi, Yukiko Matsui, Kazuya Fukushima, Sai Kudo, Kazuyuki Ban, Keisuke Kusano, Daisuke Nagano, Mami Hara, Yasuhiro Sato, Kiyoko Takatsu, Satoshi Hirata.

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
