## [Decision Letter · Decision Letter 0]

15 Jun 2020

PONE-D-20-08496

The relationship between plant-eating and hair evacuation in snow leopards (Panthera uncia)

PLOS ONE

Dear Dr. Yoshimura,

Thank you for submitting your manuscript to PLOS ONE. After careful consideration, we feel that it has merit but does not fully meet PLOS ONE’s publication criteria as it currently stands. Therefore, we invite you to submit a revised version of the manuscript that addresses the points raised during the review process.

We look forward to receiving your revised manuscript.

Kind regards,

Bi-Song Yue, Ph.D

Academic Editor

PLOS ONE

"All management for the snow leopards were in accordance with the Code of Ethics of the Japanese Association of Zoos and Aquariums, and the management guideline for the snow leopards by each institution. All sampling procedure was noninvasive for animals and approved by each institution This study was conducted in compliance with the Guidelines of ethics for animal studies established by Wildlife Research Center of Kyoto University.This study complied with applicable national laws."

a. Please amend your current ethics statement to include the full name of the ethics committee that approved your specific study.

For additional information about PLOS ONE submissions requirements for ethics oversight of animal work, please refer to http://journals.plos.org/plosone/s/submission-guidelines#loc-animal-research  

Reviewers' comments:

Reviewer's Responses to Questions

**Comments to the Author**

1. Is the manuscript technically sound, and do the data support the conclusions?

Reviewer #1: Partly

Reviewer #2: Yes

2. Has the statistical analysis been performed appropriately and rigorously? 

Reviewer #1: I Don't Know

Reviewer #2: Yes

3. Have the authors made all data underlying the findings in their manuscript fully available?

Reviewer #1: Yes

Reviewer #2: Yes

4. Is the manuscript presented in an intelligible fashion and written in standard English?

Reviewer #1: Yes

Reviewer #2: Yes

5. Review Comments to the Author

Reviewer #1: After reviewing your article, "The relationship between plant-eating and hair evacuation in snow leopards (Panthera uncia)," I have formulated the following comments and suggested modifications to the manuscript.

Major revisions:

1) Figure legends appear to be absent and would be very much welcome to help explain the data.

2) The way the data are represented in Figure 2 is a bit confusing; is there a clearer way to graph the relationship (or lack thereof) between the masses of hair and plant material?

3) A lack of correlation between plant eating and hair evacuation is fairly clearly shown, but is this a causative relationship? Please clarify whether, in this case, a lack of correlation implies a lack of causation.

Minor revisions:

1) Further proof-reading and perhaps securing the aid of a copy editor with experience editing English-language manuscripts would benefit this article.

Reviewer #2: Editorial Committee PLOS ONE

Thank you for considering my name for the revision of the manuscript: The relationship between plant-eating and hair evacuation in snow leopards (Panthera uncia) by Hiroto Yoshimura et al.

First at all I would like to congratulate the authors for this very valuable study on snow leopards feeding behavior, that will definitely add important elements to the understanding of the species ecology and provides management elements for both in situ and ex situ conservation scenarios.

In this paper the authors analyzed plant material and hair amount in scat samples of 13 (11) snow leopards (7 females, 6 males) kept in zoos in Japan, in order to test if plant consumption facilitates or it is associated with i) vomiting and ii) hair evacuation. The authors used a GLM to analyze their data and concluded that although plant consumption is common among snow leopards, there is no relationship between the ingestion of plant material and vomiting for hair evacuation. The manuscript is professionally written, and information is presented in a highly organized manner.

Due to the importance of the species as an icon of conservation and its condition as an endangered species, I recommend considering this research for editorial space in the journal.

However, several aspects need to be addressed by the authors prior publication. My major concerns are referred to the introductory and discussion sections. A greater effort needs to be made on contextualizing:

1) The threaten category of the species. I could not find a single comment on major threats facing by the species in the wild not even the UICN category was cited in the text. The authors, major players on in situ conservation efforts, should include in an explicit way, recommendations that contribute to the management of the species in captivity derived from the results of the present study. This also works for ex situ conservation efforts. The discussion section would be enriched by a comment of the authors on the applicability of results of this research in wildlife management.

2) On the same line of ideas, it would be great for the authors to share their appreciation on the importance of conducting research taking advantage of captive animals at zoos.

Although the authors make a great effort in providing information on the existing hypotheses on carnivore plant consumption, little information is provided on what kind of determinants are potentially related with plant consumption by snow leopards in their natural environments.

I fully understand that, as in many other wild felid species, the collection of information on diet for Panthera uncia is difficult in the wild. I highly recommend the authors to include some information on the general climate determinants across snow leopards´ distribution, to account for both plant phenology and animal prey availability throughout the year. It would also be important to know: i) if plants found in snow leopards scats are common or not in the wild; ii) if they coexist with animal preys used by the cats, or if the cats must move to other areas to get them; and finally, iii) what we know about the pattern of spatial distribution of these plants (aggregated, scattered, other), including a brief description on their elevational limits. These comments are relevant since Wegge et al. (2012) reported the presence of only plants in scats of snow leopards; work cited by the authors.

Wegge P, Shrestha R, Flagstad O. Snow leopard Panther uncia predation on livestock and 359 wild prey in a mountain valley in Northern Nepal: implications for conservation management. 360 Wildlife Biol. 2012;18(2):131–41.

Besides these general aspects, it would be highly appreciated if the authors will address this punctual aspects:

• Since there are several and quite different evolutionary paths among felids, the lack of sugar testing in domestic cats is an ambiguous indication on the lack of importance of plants as a nutritious resource for felids. Independently of tasting sugar, plant resources could be selected for their use based on another type of clues such as olfactory clues.

• Line 97: replace the word calve, by cubs.

• Line 334, capital letter for the name of the genus in Panthera unica.

• AGE OF THE ANIMALS: There is not information on animals age, if it is not possible to determining age, please explain why it was not taking into account?

• EXCLUSION OF WOOD MATTER: The explanation on why wood was not included as plant material can be enriched, particularly if one of the hypothesis mentioned in the introductory section claims physical properties of fibers from plants help the elimination of hair material through the digestive tract.

• Eliminate the word “of”, and the plural in the word amount, on line 210: The scatter plot shows the relationship between of the amount of hair and plant included in scat…

• Authors mention: This indicates that plant-eating behavior is common in captive snow leopards and appears even if they are not starving… First, I consider that the relationship established is not causal, lines 250.251. Based on the text, none of the animals were observed under starving conditions

6. PLOS authors have the option to publish the peer review history of their article (what does this mean?). If published, this will include your full peer review and any attached files.

Reviewer #1: No

Reviewer #2: Yes: HUGO MANTILLA-MELUK PHD

---

## [Author Response · Author response to Decision Letter 0]

9 Jul 2020

Reply to Associate Editor: Dr. Bi-song Yue

Thank you for the helpful comments provided by you and your team. We have followed all the suggestions and modified the manuscript. We hope these changes are satisfactory and sufficient.

 Thank you for your comments. We checked our manuscript and added corresponding author’s initials in parentheses after the email address on the title page (lines 21-22). 

"All management for the snow leopards were in accordance with the Code of Ethics of the Japanese Association of Zoos and Aquariums, and the management guideline for the snow leopards by each institution. All sampling procedure was noninvasive for animals and approved by each institution This study was conducted in compliance with the Guidelines of ethics for animal studies established by Wildlife Research Center of Kyoto University.This study complied with applicable national laws."

a. Please amend your current ethics statement to include the full name of the ethics committee that approved your specific study.

Thank you for pointing this out. We included the full name of the ethics committee; Animal Experimentation Committee of Wildlife Research Center of Kyoto University (line 91). 

Reply to reviewers:

Thank you for your suggestions and comments.

Our responses to the comments are shown below. In the revised manuscript, all changes are highlighted in red.

Reviewer #1: (anonymous)

Major revisions:

1) Figure legends appear to be absent and would be very much welcome to help explain the data.

Thank you for pointing this out. We added the legend for Fig. 1 (lines 212-213).

2) The way the data are represented in Figure 2 is a bit confusing; is there a clearer way to graph the relationship (or lack thereof) between the masses of hair and plant material?

Thank you for your suggestion. We modified Fig. 2 accordingly: showing data form each individual separately. The difference in sampling period for each individual is shown using different colors (white, black, or gray).

3) A lack of correlation between plant eating and hair evacuation is fairly clearly shown, but is this a causative relationship? Please clarify whether, in this case, a lack of correlation implies a lack of causation.

Thank you for pointing out this concern. Our results cannot rule out a causative relationship between the amount of plant in scat and hair evacuation. To establish this, we would require within-individual comparisons (e.g., with plants and without plants under the same captive conditions), which were difficult to procure, as we would need to remove all plants from the leopards’ enclosure to effectively ensure no plant eating. 

To this effect, we have included the following sentence on lines 289-291.

“Although we cannot rule out a causative relationship between the amount of plant in scat and hair evacuation, this study was the first to provide evidence that plant matter in scat had no quantitative effect on hair evacuation.”

Minor revisions:

1) Further proof-reading and perhaps securing the aid of a copy editor with experience editing English-language manuscripts would benefit this article.

Thank you for your suggestion. We added references 13, 14, 18, 20, 21, 22, 38, and 43 to the reference list. We also asked an experienced English-language editor to review our draft.

Reviewer #2: (Dr. Hugo Mantilla-Meluk)

First at all I would like to congratulate the authors for this very valuable study on snow leopards feeding behavior, that will definitely add important elements to the understanding of the species ecology and provides management elements for both in situ and ex situ conservation scenarios.

In this paper the authors analyzed plant material and hair amount in scat samples of 13 (11) snow leopards (7 females, 6 males) kept in zoos in Japan, in order to test if plant consumption facilitates or it is associated with i) vomiting and ii) hair evacuation. The authors used a GLM to analyze their data and concluded that although plant consumption is common among snow leopards, there is no relationship between the ingestion of plant material and vomiting for hair evacuation. The manuscript is professionally written, and information is presented in a highly organized manner.

Due to the importance of the species as an icon of conservation and its condition as an endangered species, I recommend considering this research for editorial space in the journal.

However, several aspects need to be addressed by the authors prior publication. My major concerns are referred to the introductory and discussion sections. A greater effort needs to be made on contextualizing:

(1) The threaten category of the species. I could not find a single comment on major threats facing by the species in the wild not even the UICN category was cited in the text. The authors, major players on in situ conservation efforts, should include in an explicit way, recommendations that contribute to the management of the species in captivity derived from the results of the present study. This also works for ex situ conservation efforts. The discussion section would be enriched by a comment of the authors on the applicability of results of this research in wildlife management.

 Thank you for your suggestion. We have added information on their conservation status in lines 67-69. We have also added certain discussion points about the contribution to management and enrichment in captivity in lines 255-259. Our results indicate plant-eating is normal behavior for snow leopards, hence growing plants in the enclosure will bring out their natural behaviour.

(2) On the same line of ideas, it would be great for the authors to share their appreciation on the importance of conducting research taking advantage of captive animals at zoos.

Thank you for your suggestion about the importance of conducting research in captivity. We have included this in the discussion in lines 264-269, as shown below.

“In the wild, snow leopards move daily across long distances [21] (e.g., 12 km/day in Mongolia [38]). Their broad home range and rugged habitat make it difficult to observe their natural behavior and hinder our ability to consistently collect scat samples from the same individual. Studies on captive individuals have enabled more detailed observations of this plant-eating behavior, allowing continuous sampling to test this relationship with a time difference. This study further demonstrates the importance of studying captive individuals to understand wildlife.”

Although the authors make a great effort in providing information on the existing hypotheses on carnivore plant consumption, little information is provided on what kind of determinants are potentially related with plant consumption by snow leopards in their natural environments.

Thank you for your suggestion. Unfortunately, to our knowledge, researchers are yet to investigate the driving factors of plant consumption in carnivores in natural environments. We have clarified this point and suggested this would be a suitable future area of research in lines 309-313 in the revised manuscript, as shown below:

“Another area of research that requires further investigation would be to identify the driving factors of carnivore plant consumption in the wild. Although carnivores are known to be indifferent to sugars, as demonstrated by a study on domestic cats that revealed a lack of sweet taste receptors [42], other flavors (e.g., bitter taste), olfactory clues [43], or plant texture [8] might be influencing carnivore plant-eating behaviour. ”

 I highly recommend the authors to include some information on the general climate determinants across snow leopards´ distribution, to account for both plant phenology and animal prey availability throughout the year.

Thank you for your suggestion. Snow leopards are widely distributed in 12 countries across Central Asia, making it difficult to uniformly describe the detailed characteristics of climatic variables such as temperature and precipitation. Instead, we added some information about their habitat and vegetation in lines 71-73, as shown below.

“Large portions of snow leopards’ natural habitat are devoid of tree cover, given the predominance of alpine and desertic climate conditions in their natural range. The vegetation in their range varies from scrubland and desert to forest-alpine ecotones”.

 It would also be important to know:

 i) if plants found in snow leopards scats are common or not in the wild;

Thank you for your suggestion. It is considered to be common in the wild. We have provided a reference to this effect from Wegge et al. (2012) in lines 75-77.

ii) if they coexist with animal preys used by the cats, or if the cats must move to other areas to get them; 

Thank you for your suggestion. We have added information on typical prey items and their respective range in lines 69-71.

and finally, iii) what we know about the pattern of spatial distribution of these plants (aggregated, scattered, other), including a brief description on their elevational limits. These comments are relevant since Wegge et al. (2012) reported the presence of only plants in scats of snow leopards; work cited by the authors.

Wegge P, Shrestha R, Flagstad O. Snow leopard Panther uncia predation on livestock and 359 wild prey in a mountain valley in Northern Nepal: implications for conservation management. 360 Wildlife Biol. 2012;18(2):131–41.

Thank you for your recommendation. Snow leopards are widely distributed through a broad range in Central Asia, across which vegetation survey data are scarce. This lack of data across a geographically and physically varied region makes it difficult to provide specific altitudinal limits for plants. Thus, we added general information about snow leopards’ typical habitat and vegetation in lines 71-73.

Besides these general aspects, it would be highly appreciated if the authors will address this punctual aspects:

Since there are several and quite different evolutionary paths among felids, the lack of sugar testing in domestic cats is an ambiguous indication on the lack of importance of plants as a nutritious resource for felids. Independently of tasting sugar, plant resources could be selected for their use based on another type of clues such as olfactory clues.

Thank you for your suggestion. We have deleted the sentence on taste receptors and added the fact that felids are known to frequently eat grass and leaves, which should be less nutritional than fruits, in lines 57-58.

• Line 97: replace the word calve, by cubs.

Thank you for pointing out. We changed the word to “cubs” (line 99).

• Line 334, capital letter for the name of the genus in Panthera unica.

Thank you for pointing out this mistake. We changed it to a capital letter (line 386).

• AGE OF THE ANIMALS: There is not information on animals age, if it is not possible to determining age, please explain why it was not taking into account?

Thank you for your question. We provided data about the animals’ age in our supporting information, as a table in our original submission. We moved this table into the main body of the manuscript, it is now Table 1 in the revised manuscript.

• EXCLUSION OF WOOD MATTER: The explanation on why wood was not included as plant material can be enriched, particularly if one of the hypothesis mentioned in the introductory section claims physical properties of fibers from plants help the elimination of hair material through the digestive tract.

Thank you for your suggestion. We first removed wooden chips because they were not from the plants growing in the enclosure, but as you mentioned, wooden chips may work too. We, therefore, corrected the original sentence (lines 129-130, line 150) and re-analyzed the data with wood matter included. Biting wooden structures was additionally counted as “plant eat” in Table 2. As for scat samples, four samples from Male 6 contained wooden matter, so this individual was included in the analysis (lines 207-208, line 228). Samples from Female 7 did not contain any plant matter, thus were excluded from further analysis (lines 226-227). Despite these new results, our overall conclusions remain the same. Changed values are shown in Table 2, 3 and 4.

• Eliminate the word “of”, and the plural in the word amount, on line 210: The scatter plot shows the relationship between of the amount of hair and plant included in scat…

Thank you for pointing this out. We deleted the word “of” (line 223).

• Authors mention: This indicates that plant-eating behavior is common in captive snow leopards and appears even if they are not starving… First, I consider that the relationship established is not causal, lines 250.251. Based on the text, none of the animals were observed under starving conditions

Thank you for your comment. The discussion was related to a previous study discussing prolonged starvation as the driver for plant-eating in leopards (Panthera pardus). We have changed the order of sentences in lines 270-274 to clarify this connection.

---

## [Editor Report · Decision Letter 1]

13 Jul 2020

The relationship between plant-eating and hair evacuation in snow leopards (Panthera uncia)

PONE-D-20-08496R1

Dear Dr. Yoshimura,

We’re pleased to inform you that your manuscript has been judged scientifically suitable for publication and will be formally accepted for publication once it meets all outstanding technical requirements.

Kind regards,

Bi-Song Yue, Ph.D

Academic Editor

PLOS ONE

---

## [Editor Report · Acceptance letter]

17 Jul 2020

PONE-D-20-08496R1 

The relationship between plant-eating and hair evacuation in snow leopards (Panthera uncia) 

Dear Dr. Yoshimura:

I'm pleased to inform you that your manuscript has been deemed suitable for publication in PLOS ONE. Congratulations! Your manuscript is now with our production department. 

Kind regards, 

on behalf of

Dr. Bi-Song Yue 

Academic Editor

PLOS ONE